# Evaluating User Perceptions of a Vibrotactile Feedback System in Trunk Stabilization Exercises: A Feasibility Study

**DOI:** 10.3390/s24041134

**Published:** 2024-02-09

**Authors:** Philipp Floessel, Lisa-Marie Lüneburg, Julia Schneider, Nora Pohnert, Justin Foerster, Franz Kappert, Doris Lachmann, Jens Krzywinski, Uwe Platz, Alexander Carl Disch

**Affiliations:** 1Center of Orthopedic, Trauma and Plastic Surgery, University Hospital Carl Gustav Carus, Medical Faculty Carl Gustav Carus, Technische Universität Dresden, 01307 Dresden, Germany; justin.foerster@ukdd.de (J.F.); doris.lachmann@ukdd.de (D.L.); uwe.platz@uniklinikum-dresden.de (U.P.); alexander.disch@uniklinikum-dresden.de (A.C.D.); 2Industrial Design Engineering, Faculty of Mechanical Engineering, Technische Universität Dresden, 01219 Dresden, Germany; lisa-marie.lueneburg@tu-dresden.de (L.-M.L.); franz.kappert@ukdd.de (F.K.); jens.krzywinski@tu-dresden.de (J.K.); 3Universitäts-Physiotherapie-Zentrum, University Hospital Carl Gustav Carus, Medical Faculty Carl Gustav Carus, Technische Universität Dresden, 01307 Dresden, Germany; nora.pohnert@ukdd.de; 4University Comprehensive Spine Center, University Hospital Carl Gustav Carus, Medical Faculty Carl Gustav Carus, Technische Universität Dresden, 01307 Dresden, Germany

**Keywords:** vibrotactile feedback system, trunk stability, NASA TLX questionnaire

## Abstract

Low back pain patients often have deficits in trunk stability. For this reason, many patients receive physiotherapy treatment, which represents an enormous socio-economic burden. Training at home could reduce these costs. The problem here is the lack of correction of the exercise execution. Therefore, this feasibility study investigates the applicability of a vibrotactile-controlled feedback system for trunk stabilisation exercises. A sample of 13 healthy adults performed three trunk stabilisation exercises. Exercise performance was corrected by physiotherapists using vibrotactile feedback. The NASA TLX questionnaire was used to assess the practicability of the vibrotactile feedback. The NASA TLX questionnaire shows a very low global workload 40.2 [29.3; 46.5]. The quality of feedback perception was perceived as good by the subjects, varying between 69.2% (anterior hip) and 92.3% (lower back). 80.8% rated the feedback as helpful for their training. On the expert side, the results show a high rating of movement quality. The positive evaluations of the physiotherapists and the participants on using the vibrotactile feedback system indicate that such a system can reduce the trainees fear of independent training and support the users in their training. This could increase training adherence and long-term success.

## 1. Introduction

With a lifetime prevalence of 80 to 85%, low back pain is one of the most common musculoskeletal disorders worldwide [1,2,3]. The cost burden on the healthcare system is expected to continue to increase in the coming decades [4,5]. To counteract this trend, early identification of patients at risk for back pain is of general importance, as is the use of effective primary preventive measures to avoid Low back pain (LBP) [1,6]. It has been shown that well-developed trunk stability is crucial for this purpose, both in elite sports and in the general population [7,8,9,10]. Recent studies show that trunk stabilising sensorimotor training performed twice a week for twelve weeks can reduce recurring rates in LBP patients by 80% [11,12]. To this end, most patients with LBP are treated in physiotherapy practices [13,14]. Most people use an outpatient rehabilitation facility for this purpose. However, physiotherapy treatment is one of the most staff-intensive areas of medical care [15,16]. On the other hand, home-based training is an alternative form of treatment that can take place in the rehabilitant’s familiar surroundings. It facilitates the transfer of treatment into everyday life. In this context, independent home practice is becoming increasingly important. The costs for the same rehabilitation success can be reduced by about 50% through independent intensive practice at home [17].

### 1.1. Theory

According to recent studies in the field of the post-operation treatments, an independent but also significantly intensified home practice achieves the same rehabilitation success as guided physiotherapy [18]. The patient must integrate the rehabilitation measure into their daily routine. The success of home training depends decisively on individual discipline, self-motivation and the correct execution of the exercises by the patient. The key to success lies in the combination of professional treatment by the physiotherapist and independent intensification of therapy at home. A high quality of movement is essential for a successful rehabilitation. In professional treatments, this can be achieved by therapist feedback directly to patients through touch or verbal communication. For home-based training, however, no physiotherapists are present. Therefore, a different approach must be applied to ensure the necessary movement quality.

### 1.2. Vibrotactile Feedback/Tactile Internet

The tactile internet opens up technical possibilities for providing feedback. The almost latency-free data transmission allows data to be measured and real-time feedback to be transmitted [19]. A vibrotactile-controlled feedback system is a promising variant that enables autonomous training, but also monitors training quality and supports the trainee in performing exercises correctly [20]. The tactile feedback is perceived via the skin. The information can be generated by pressure or vibration, with the latter being referred to as vibrotactile feedback [21].

Tactile feedback can be given directly to patients by therapists. In a hands-off context, a human-machine interface is required to provide tactile feedback. Feedback is usually provided by small, lightweight, vibrating transducers, also known as actuators.

Vibrotactile feedback can be positioned in different locations on the body and in different ways to respond to the stimulus [22,23,24,25]. Users are usually instructed to move in the direction of the stimulus (attractive cue) [22,26]. In a comparative study between attractive or repulsive feedback, Kinnaird et al. [23] showed that repulsive feedback was superior to attractive feedback in healthy older adults on the lower back. However, further studies show that the use of attractive feedback can be just as effective in correcting motor tasks [23,27].

In this context, it should also be borne in mind that attractive feedback may be cognitively associated with a reward and therefore has a more positive connotation in the user’s evaluation and could therefore increase the trainee’s motivation and ultimately training compliance. In contrast, repulsive feedback could be interpreted as an alarm, which could be associated with negative emotions, as it suggests to the exerciser that they are not performing the exercise correctly. In addition, Ma and Lee [24] were able to show that attractive stimulation, without further instructions, causes the subjects to align their body position with the attractive stimuli. Ma and Lee [24] conclude that an improved internal proprioceptive representation and orientation is responsible for this. By utilizing this natural tendency, the attractive feedback could be used just as well as the repulsive feedback [24].

However, research as well as the use of vibrotactile feedback devices in a therapeutic context is still insufficient, so that little can be said about the design of technical feedback systems [20]. The aim of this research project is therefore to derive requirements for the design of a tactile feedback strategy, especially for the implementation of trunk-stabilising exercises. To this end, this study examines the feedback behaviour of healthy people and analyses their use of technical tactile feedback.

From this point of view the following questions arise: is it possible to correct healthy adults using vibrotactile feedback while performing trunk-stabilising exercises and to improve the quality of movement compared to an uncorrected exercise? These two research questions can be generated from these questions:

RQ1: “Is it possible to use vibrotactile feedback to support people in correctly performing physiotherapeutic exercises?”

RQ2: “What requirements do users have for vibrotactile technology-based feedback to accept it?”

## 2. Materials and Methods

### 2.1. Sample

The sample consisted of 9 women and 4 men with an average age of 46.5 years (SD ± 9.6), an average height of 173.4 cm (SD ± 10.0), an average weight of 72.7 kg (SD ± 8.4) and a BMI of 24.0 (SD ± 1.8). The recruitment and examination of the subjects took place at a DOSB-licensed sports medicine facility. Inclusion and exclusion criteria are listed in Table 1.

To enable vibrotactile feedback for the test subjects, they wore a tight-fitting long-sleeved top with twelve vibration modules that generated vibrations in the frequency range of 10–400 Hz using ERM vibration motors. The prototype also includes three circuit boards and the cabling for the components.

All components are shielded from the test subject by housings. All electronic components are insulated and the applied voltage of 5 V and a current of 0.9 A pose no danger to healthy test subjects in the event of a defect. Power was supplied via a USB cable connected to a laptop. The ERM vibration motors were localized to the different areas of the body (Figure 1).

### 2.2. Materials

The applicability of vibrotactile feedback as a tool for correcting exercise performance was evaluated by both the physiotherapist and the subjects. The movement quality as well as the assessment of the feedback quality of the subjects regarding the vibrotactile feedback are carried out by means of standardised questionnaires. The questionnaire for physiotherapists was not validated in prior studies.

### 2.3. Physiotherapist Questionnaire

The feedback giving physiotherapists gave a third-party assessment on different variables. In order to assess the specific functionality of the vibrotactile feedback to support a correct execution of trunk-stabilising exercises, the correction behaviour by physiotherapists was evaluated on a 4-point Likert scale (poles: not at all, a little, moderately, very):(1)Ability to implement the vibrotactile feedback(2)Level of difficulty for the subject to perform the exercises(3)Accuracy of axis and thus the quality of movement of the subjects

While the subjects performed the exercises, the exercise execution was controlled by the physiotherapists using vibrotactile feedback. If an exercise was performed incorrectly, the physiotherapists triggered vibrotactile feedback at the relevant part of the body. The test subjects were asked to react as appropriately as possible by correcting the movement. In order to assess the applicability of the vibrotactile feedback, the physiotherapists were asked to evaluate the subjects’ corrective behaviour after the exercise. For this purpose, the physiotherapists were asked the following three questions:(1)How would you rate the subjects’ ability to implement the correction instructions using the vibrotactile feedback given?(2)How would you rate the difficulty of the exercises?(3)How do you assess the axial accuracy (hip-upper body) of the movement execution and thus the movement quality of the subjects before and after the given correction instructions by means of the given vibrotactile feedback?

The physiotherapists were given a 4-point Likert scale to answer each of the three questions (poles: (1) not at all, (2) a little, (3) moderately, (4) very much).

The description of the data is based on the median and the quartiles 0.25 and 0.75.

### 2.4. Subject Questionnaire

The subject questionnaire focused on capturing subjective user perceptions and eliciting potential users’ requirements for vibrotactile technology-based feedback. The NASA TLX was used for this purpose [28], which can be used to assess psychological workload. It provides a global rating of subjective workload. In addition to the global rating, six subscales can be calculated (mental, physical and temporal demand, effort, performance, frustration level).

The mental demand scale refers to the mental and perceptual activity a user needs to complete a task. This measure includes mental activities such as thinking, decision making, remembering, looking and similar aspects. High levels indicate that a task was demanding and complex, whereas low levels are indicators for simple and easy tasks.

Physical demand refers to actual physical efforts needed by users for task completion. It includes for example activities like pushing, pulling and such. High levels on this subscale indicate that a task was demanding, brisk or laborious. Low levels indicate an easier and slack task with possibilities for slow execution or even resting.

Effort means the amount of work users put into the completion of a task. It combines the ratings of physical and mental activities during the task execution and therefore refers to a rating of overall strain or struggle.

The scale temporal demand measures the subjective time pressure users feel during a task execution. It depends on the pace of task elements. In this case, high outcome refers to frantic or rapid pace, whereas low outcomes indicate the subjective time perception as more slow and leisurely.

The rating of performance indicates a self-rating on task success by the users in terms of goal completion. It includes the users’ satisfaction with the accomplishment.

Frustration level refers to certain feelings a user may have while performing a task. High levels indicate that they are insecure, discouraged, irritated, stressed or annoyed during a task, while low levels indicate opposite aspects such as feeling gratified, content, relaxed and complacent.

In addition to the NASA TLX, subjects were asked to rate the intensity of the feedback related to the location of the feedback indicator in different areas of the body. This measure was imposed to find out whether users are able to feel the vibro-tactile at all while performing physically demanding tasks. Feedback intensity was rated on a 4-point Likert scale (anchors: not noticeable at all, weak, good, too strong). To ascertain whether the feedback can be associated with an intended purpose, subjects were asked about their subjective degree of approval on six potential purpose associations (alarming, warning, indicative, leading, supportive, meaningless) using a 2-point Likert Scale (anchors: does apply, does not apply). To rule out the idea that future users may have negative associations regarding the new system, a range of feedback-associated emotions were rated by the participants on the same scale as purpose.

### 2.5. Study Procedure

To provide vibrotactile feedback for the test subjects, they wore a tight-fitting long-sleeved top with twelve vibration modules. Three different types of exercises were used during the study, to on the one hand exclude effects regarding specific kinds of exercises, and on the other hand to ensure that subjects can encounter the need for a correction. The exercise portfolio consisted of the quadruped stand with diagonal arm-leg coordination. The second exercise was to train the shoulder blade fixators while leaning against a wall in a two-legged stand by moving the arms up and down. The third exercise was to present a lateral support with pelvic raise and lowering (see Figure 2). All exercises were performed on an unstable surface in order to particularly stress the neuromuscular control function. The three basic exercises were practised with the subjects. Correct execution and error patterns were shown to them in detail by a physiotherapist and described verbally before they began to perform the exercises independently. Basically, the subjects were asked to move in the opposite direction of the stimulus from the feedback (repulsive cue). Only in the scapula fixator exercise were the subjects asked to move in the direction of the stimulus (attractive stimulus) when lowering the shoulders: M. (trapecius pars ascendens). The same applied when tensing the arms backwards (middle third Os humeri/Humerus). The subjects were asked to perform ten repetitions of each of the three exercises. The exercise performance was corrected by a physiotherapist using vibrotactile feedback. As the exercise quality was to be assessed independently by two physiotherapists, the subjects completed two rounds, meaning each exercise was performed two times.

The presentation of the exercises took place in a laboratory in the area of the DOSB-licensed sports medicine facility.

The physiotherapists were in the same room as the subjects while they were correcting them via vibrotactile feedback. However, they did not have any eye contact. During this session, the physiotherapist also gave no verbal feedback. The subjects were able to concentrate fully on the vibrotactile feedback while performing the exercise. Both physiotherapists were experienced in using the devices and providing vibrotactile feedback. They were significantly involved in the six-month development phase.

## 3. Results

### 3.1. Research Question 1

The first research question was related to the general functionality of the feedback: “is it possible to use vibrotactile feedback to help people perform physiotherapeutic exercises correctly?” To answer the question, the subjects’ quality of movement after correction and feasibility of feedback by physiotherapists, as well as the workload during exercise performance by the subjects were collected.

### 3.2. Quality of Movement and Feasibility

As a general starting point, it was important to determine whether the vibrotactile technology-assisted feedback can have a general and positive effect on the execution of an exercise, as well as whether the subjects can implement the feedback (“Feasibility of vibrotactile feedback”). The results showed an overall high evaluation of the quality of movement in the corrected subjects 3 [3; 3]; range between 1–4, with 4 being the best). This indicates the general functionality as well as the quality of the feedback technique in a physiotherapeutic setting. None of the physiotherapists chose the anchor “not at all”.

Furthermore, the physiotherapists were asked whether the subjects were able to implement the feedback in their exercises during the study. The results on implementability indicate an overall good rating of the variable “quality of movement” (3 [3; 3]; range between 1–4, with 4 being the best). No physiotherapist chose the option “not at all”. In general, the subjects were able to integrate the feedback into their movements.

The degree of “difficulty of the entire exercise program” was rated by the physiotherapists as (3 [3; 3]; range between 1–4, with 4 being the best).

### 3.3. Workload

To determine whether the cognitive processing of the technology-assisted vibrotactile feedback interferes with users performing the exercises, the unweighted NASA TLX was evaluated. The global workload was 40.2 [29.3; 46.5], indicating a very low workload. To examine the distribution of mental workload in more detail, scores were calculated for each of the NASA TLX subscales. The result for the subscale on mental workload can be classified as a medium value: 46.6 [26.3; 64.8]. The highest scores were obtained for the subscale of physical demand: 56.8 [36.1; 63.2] The result for the subscale of temporal demand is very low 18.6 [15.1; 21.7]. The results of the performance subscale reach a medium value: 31.8 [18.9; 36.3]. The results of the subscale effort reach a middle value: 56.6 [25.8; 81.5]. Frustration subscale also reached a low level 16.5 [11.9; 21.9]; the global workload and all subscales are shown in Figure 3.

### 3.4. Research Question 2

The second research question was related to technology acceptance: “what requirements do users have for vibrotactile technology-based feedback to be acceptable?” To answer this question, the subjective evaluations of the feedback in terms of intensity, purpose and the associated positive or negative feelings from the users’ perspective were surveyed.

### 3.5. Intensity of the Vibration

An essential question in connection with this second research question is whether the vibrotactile feedback during the execution of an exercise is strong enough to be perceived by the user. To answer this question, the subjects gave subjective assessments of the intensity of the vibration in relation to the different areas of the body where feedback could be given. The anchors “not noticeable at all” and “weak” were combined into one category to represent the area identified as “needing improvement”. The anchor “too strong” was not selected by any of the participants.

The assessment was made on a 4-point Likert scale. For the subsequent interpretation of the data, the anchors were assigned the numbers 1–4: 1 = “not noticeable at all”; 2 = “weak”, 3 = “good”, 4 = “too strong”.

Overall, the strength of the vibration was perceived well in all areas of the body. The areas that were perceived least well were the abdominal area and the hip area (anterior/posterior). A total of four test subjects rated the feedback in the abdominal area as “too weak”. In the hip area, three people rated the feedback as “too weak” or “not noticeable at all”.

In the neck and shoulder areas, only two people rated the anchor as “too weak”. The best feedback was perceived in the lower back area and on the arms. In each of these two areas, only one test subject rated the feedback as “too weak”.

The detailed results for the different body areas are listed in Figure 4.

## 4. Discussion

The purpose of this feasibility study was to investigate whether this technology-based feedback approach is suitable as a correction tool for the execution of trunk-stabilising exercises and how it is perceived. In this context it was examined to which extent the subjects were able to correct their execution in real time in response to given vibrotactile impulse. The vast majority of the test persons rated the feedback as helpful for the implementation of the exercises. The physiotherapists also rated independently of each other that the test persons were able to implement the vibrotactile feedback instructions initiated by them well to very well, and were accordingly able to correct the exercise execution in real time without interrupting the flow.

### 4.1. Quality of Movement and Feasibility

As a general starting point, it was important to determine whether vibrotactile technology-assisted feedback can have a general and positive effect on the exercise execution. The results showed the general functionality as well as the quality of the feedback technique in a physiotherapy setting. Based on the positive results of the physiotherapist survey, it can be deduced that the subjects were able to integrate the feedback into their movements. By adequately eliminating the movement errors and responding adequately to the vibrotactile feedback given by the physiotherapists, it can be assumed that the general quality of movement has increased. This is consistent with the findings of Nanhoe-Mahabier et al. [25] who investigated the effect of artificial vibrotactile biofeedback on trunk sway in Parkinson’s patients. In a two-group test design, where one group received balance exercises using vibrotactile feedback, Nanhoe-Mahabier et al. [25] were able to demonstrate that the group with vibrotactile feedback increased movement quality and had positive effects on trunk stability [25]. In another study related to learning complex motor tasks, Sigrist et al. [29] showed, among other things, the effectiveness of learning a rowing stroke using haptic feedback. The results showed that visual feedback in combination with haptic feedback leads to significantly better results than purely visual feedback. Sigrist et al. [21] concluded that haptic feedback can be helpful for learning complex motor tasks [29]. The decrease in heart rate can be seen as a broad argument for optimizing the quality of movement. A plausible explanation for the drop in heart rate is that the control of the muscles and the entire movement sequence becomes more efficient. This means that the body uses less energy under the same load, which is reflected in a drop in heart rate. This is how Sigrist et al. [29] demonstrated their results. In a group comparison, they were able to demonstrate a significant (*p* < 0.02) decrease in heart rate during the course of the tests during vibrotactile-supported exercises. This indicates that vibrotactile feedback allows the athletes to optimize the movement sequence of a previously learned movement, and can be helpful [29].

Furthermore, in the current study both the best and worst perceptions were attributed to repulsive feedback points. 92.3% of the subjects rated the intensity of the vibrotactile feedback on the lower back as “well noticeable” but only 69.2% in the anterior hip area which is consistent with previous studies [23].

In a crossover design study, Kinnaird et al. [23] examined the effect of attractive and repulsive vibrotactile feedback on the correction of postural control in healthy older adults. They observed a better response to repulsive cues, but the rates of change between the pre- and post-test were greater for the attractive feedback. Kinnaird et al. [23] concluded that both attractive and repulsive feedback can be used to adjust postural control [23].

Lee and Sienko [30] noted that in a comparison between attractive and repulsive instruction stimuli, a repulsive vibrotactile instruction cue leads to the greatest correlation between expert and subject movement, and is more suitable for correction than an attractive feedback strategy [30].

In contrast, attractive feedback was used for the activation exercise of the trapezius pars transversa/trapezius pars ascendens muscle. In this case, 84.6% of the subjects rated the intensity of the feedback as easily noticeable. Although this study focuses on the recording of subjective user sensations and not on measurements of vibrotactile detection, the results suggest that it is possible to use both repulsive and attractive feedback in vibrotactile feedback training of trunk-stabilising exercises. However, van Breda et al. [31] pointed out significant need for research in the thematic of the use of vibrotactile feedback in the field of motor learning [31].

### 4.2. Workload

In view of these results, Research Question 1 can be answered. It is indeed possible to initiate postural correction through technology-based vibrotactile feedback. Not only does it work in general, but it also leads to good results in postural correction with regard to exercise performance from an expert’s point of view. The amount of work required to process the feedback is not so high that the subjects are prevented from actually performing the exercises tested.

The result for mental demand can be classified as a medium value (mean 47.7). It is in an absolutely acceptable range for the task and indicates that the required mental capacities are not too loaded by the processing of the vibrotactile feedback. Users should therefore be able to use the system without too much workload.

The more complex the task, the higher the expected values. Grier [32] provides the following guideline values: simple routine tasks; 25 points, complex tasks; 42 points, creative tasks; 40 points, problem-solving tasks; 45 points, stressful tasks; 50 points [32].

The highest scores were obtained for physical demand and effort. Both aspects exceed the mean threshold on the NASA TLX subscale and they are strongly associated with physical activity, with a higher score being acceptable due to the characteristics of the exercises [32]. Nevertheless, the results for physical demand and effort are acceptable as long as they do not prevent the subjects from completing their tasks. However, Chi et al. [33] pointed out that the addition of a mental challenge combined with high-intensity physical work can influence actual performance and increase the perceived workload [33].

The result for time demand is very low, indicating that the subjects did not feel much time pressure during the exercises. Frustration also reached a low level, which indicates that the subjects did not feel very frustrated or insecure during the implementation. Both aspects are therefore well below the average threshold of the NASA TLX questionnaire [32]. These results are plausible because the use of vibrotactile feedback does not require any additional time since the correction takes place in real time during the exercises. However, Campoya Morales et al. [34] indicate that the combination of physical and mental workload results in higher values of the time dimension and thus also increases the overall score of the global workload index [34].

These results are consistent with the subjective rating of the negatively connoted feedback-associated feelings listed under associated feelings in Research Question 2, with frustration at 0%, being not chosen by any participant. The value for performance is located in the middle range of values [32]. It was expected that the expression for achievement would be between medium and high values, since getting feedback obviously means that the subject is doing something wrong that needs to be corrected. Any correction can of course be attributed to poor performance. This may diminish the rating of the degree of success on this variable. So the results show a desirable result because the score is still relatively low. This is a very positive result, because the subjects do not seem to interpret the receipt of feedback as failure, which can lead to negative emotions. In this context it should be taken into account that performance can also decrease due to under-challenging. This reduces alertness during exercise. Therefore, the good results in the area of performance could also be due to the fact that the test subjects developed increased mindfulness during the execution of the exercise by correcting based on the vibrotactile feedback, which had a positive effect on their self-assessment of performance [32,35,36].

### 4.3. Intensity of the Vibration

An essential question related to this second research question is whether the vibrotactile feedback during the execution of an exercise is strong enough to be perceived by the user. The analysed data show that the quality of feedback perception was generally perceived as good. However, the perception varied depending on the body part addressed. The vibrotactile feedback is perceived less well at positions far from the head than at positions close to the head. For example, perception at the hips and stomach is less good than at the upper body areas. This is largely in line with the results of Bao et al. [37] who used a similar approach in their study and found shorter reaction times for healthy adults at feedback points close to the head compared to feedback points far away. This could be due to the fact that the receptor density and the size of the receptive fields are related to the cortical representation and therefore the tactile spatial acuity of the skin and the density of the mechanoreceptors in different areas of the body vary greatly [37]. This means that both the perception and the neuromuscular response to vibrotactile stimuli are favoured in densely populated body regions, which are particularly close to the head [25]. Therefore, it is recommended to vary the intensity of the vibrations depending on the body region in future studies in order to achieve a similar perception for all body areas. To further optimise the feedback, it could be helpful to adjust the frequency of the feedback in addition to varying the intensity depending on body region. In this regard, the results of Stuart et al. [38] can be interpreted, who were able to determine different detection thresholds between different body regions in healthy individuals [38]. They substantiate their results with the fact that vibrotactile sensitivity depends on different groups of peripheral receptors, which can be differentiated into four different channels. These have the ability to perceive both high-frequency (60–1000 Hz) vibrations and, due to their fast-adapting afferent fibres, also low-frequencies (0.4–100 Hz). Depending on the size of the receptive fields and the density of the receptors, there are clear differences between bald skin areas and hairy skin areas with regard to vibrotactile sensitivity [38]. In addition, Morioka et al. [39] were able to observe a correlation in the perception of threshold values of vibration depending on size and location of the actuator. The effect varied with regard to proximal or distal positioning [39]. The aspect of different perceptual sensitivity depending on the location of the actuators on the body should also be taken into account when deciding whether to use attractive or repulsive feedback [27]. An additional interesting aspect of the results is that no subject chose the anchor “too strong” as an option for vibration intensity. It would be interesting for future research to generally test at what intensity a vibration is perceived as “too strong” in a realistic context and whether “too strong” even exists in certain or all areas. If this is not the case, the vibration intensity for embodied feedback technology can be further increased. This can reduce the risk of users perceiving it as too weak to almost zero. This has the potential to minimise the system’s susceptibility to error.

### 4.4. Limitations and Future Research

Some limitations could be considered. Although this was an exploratory study, no power calculation was performed to assess sample size, and a post hoc analysis might conclude that the observed power was insufficient to draw conclusions regarding the generalizability of the results. Furthermore, the unequal distribution between men and women and the small sample size of only 13 people must be taken into account as limitations.

In addition, there has so far been no use of the NASA TLX questionnaire in the literature in a sports-related, virbrotactile feedback context. This circumstance makes it difficult to interpret the individual results collected—especially the sub-shells. Another issue is that the questionnaire for physiotherapists has not been validated in previous studies.

Therefore, it is not known whether the current results from healthy people can be transferred to other target groups. Further research is needed to compare the current study results with other target groups, such as athletes or patients. This would contribute to the generalizability of the results.

Furthermore, follow-up work should investigate whether the perception of body parts such as the abdomen or hips can be improved in healthy people by adjusting the vibrotactile parameters such as intensity or frequency. Furthermore, it should be tested whether back pain patients react similarly to healthy persons to the vibrotactile feedback or whether the feedback must be fundamentally changed in this cohort. This also concerns the positioning of the sensors in pain patients. Here, it must be checked which distance from the pain point must be maintained in order not to increase the pain due to the haptic feedback but nevertheless to achieve the desired movement correction.

In order to increase the trainees’ motivation, it could be considered in the future to underpin the vibrotactile feedback with an acoustic feedback which motivates the user with positive statements when the movement is performed more correctly. Thus, in the setting of a home-based training, this system could increase the training fidelity and the long-term success.

## 5. Conclusions

In this feasibility study, we investigated for the first time the practicability of a vibrotactile feedback system for motion guidance in the context of physiotherapy. We did not only proof the technological and clinical feasibility but also the user experience and acceptance of such a system at the same time. By that, we can show that the use of real-time feedback via a haptic waistcoat could be a promising approach to correct trunk-stabilising exercises. The results of the expert evaluation indicate a significant improvement of quality of movement through the vibrotactile feedback and can be compared with the effect of the “real” haptic feedback of a physiotherapist. In addition, the physiotherapists rated the ability of implementation as high. This has resulted in the overall assessment of the experts that such a system has a high practicability for remote active motion guidance. The results of the NASA TLX questionnaire suggest that the use of such a system can reduce the trainees’ fear of independent training and support the users instead.

## Figures and Tables

**Figure 1 sensors-24-01134-f001:**
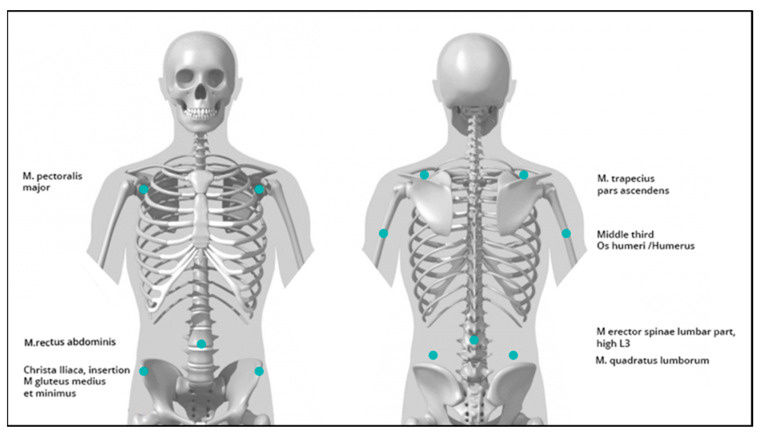
Actuator fixation of vibrotactile feedback.

**Figure 2 sensors-24-01134-f002:**
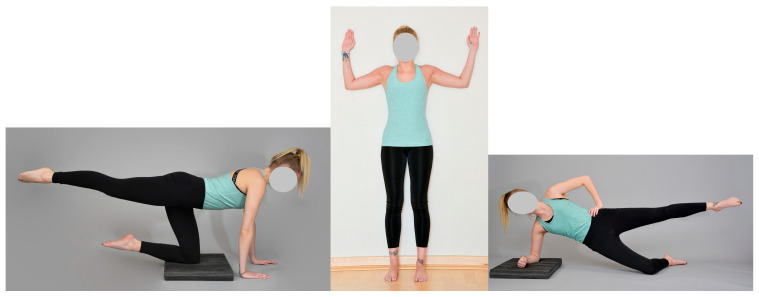
Trunk train exercise portfolio.

**Figure 3 sensors-24-01134-f003:**
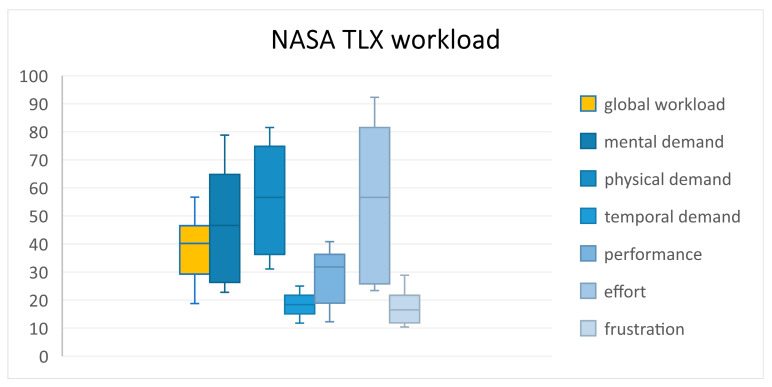
Global workload and subscales of the NASA TLX questionnaire.

**Figure 4 sensors-24-01134-f004:**
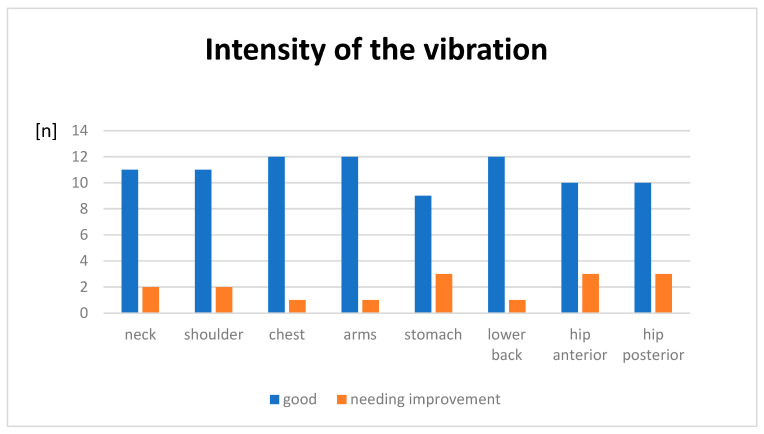
Feedback perception on different body region.

**Table 1 sensors-24-01134-t001:** Inclusion and exclusion criteria for individuals to be included in the study.

Inclusion Criteria	Exclusion Criteria
Back pain free > 3 months	acute back pain
40 and 60 years of age	Pregnancy
Subject can stand on one leg	Dizziness
	Ankle, knee, hip joint injuries within last three months
	postoperative < 6 months
	unable to support themselves on their hand or knee joints

## Data Availability

Data are contained within the article.

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
