# Peer review of "Evaluating User Perceptions of a Vibrotactile Feedback System in Trunk Stabilization Exercises: A Feasibility Study"

_sensors, 2024, doi:10.3390/s24041134_

Round 1

Reviewer 1 Report

Comments and Suggestions for Authors

I appreciate the opportunity provided by the authors and editors to review the manuscript titled “Use of Vibrotactile Feedback to Improve Execution Quality of Core Stability Exercises.” This study explores the efficacy of a vibrotactile-controlled feedback system in trunk stabilization exercises, tested on 13 healthy adults performing three types of exercises. It primarily focuses on assessing users' perceptions through self-administered questionnaires and reports predominantly positive feedback. While the study presents interesting insights and addresses a significant research objective, there are several crucial aspects that require further attention and clarification.

General comments

1.       The manuscript needs a clearly stated research question, ideally framed using the PICO format.

2.       The authors assert that the feedback system improves core stability exercise quality, yet the manuscript lacks objective data on movement quality. This data is essential for validating the research objective.

3.       The sample size of 13 is notably small. Please justify this choice. Additionally, the absence of statistical tests hinders the generalizability of the results. A rationale for this methodological choice is necessary.

Specific points

Title:

Ÿ   Consider revising the title to better reflect the manuscript's content. A suggestion is: “Evaluating User Perceptions of a Vibrotactile Feedback System in Trunk Stabilization Exercises.”

Abstract:

Ÿ   The abstract should explicitly state the research question.

Introduction:

1.       Line 40-41, Page 1: Please cite a source for the claim made here.

2.       Lines 72-77, Page 2: Clarify and standardize the use of terms “repulsive” and “reimpulsive.”

3.       Lines 78-79, Page 2: A reference is needed for the statement made here.

Materials and Methods:

1.         Justify the small sample size selection.

2.         Clarify whether physiotherapists were present during the exercises and discuss the implications of this on the independence of the training.

3.         Detail the physiotherapists’ familiarity with the device and their experience with assistive technology.

4.         Include the total time required for each training session, encompassing preparation, execution, and cleanup. Also, address the effort needed for these phases from both participants' and physiotherapists' perspectives.

5.         If the questionnaire for physiotherapists wasn't validated in prior studies, this should be noted in the Methods and Discussion sections.

6.         Specify the location of the exercises (participant’s home or a lab).

 Results:

1.       Provide data on the changes in movement quality over time during the training.

2.       For scales, like the 4-point Likert scale, use median and IQR instead of mean and SD.

3.       Justify the lack of statistical tests to support the generalizability of the findings.

 Discussion:

1.       Discuss potential biases in the study.

2.       Elaborate on how the results can be generalized to the target population, clarifying who this population is.

3.       Discuss the extent to which the training improves movement quality and compare its effectiveness with and without the device.

4.       Clearly outline the study's limitations.

Author Response

Ladies and gentlemen,

thank you very much for reviewing our manuscript.
We have incorporated your suggestions for changes in the manuscript. The attachment can be found in the appendix.

Reviewer 2 Report

Comments and Suggestions for Authors

The subject matter of this study is very interesting and opens up new lines of work and feedback for the physiotherapy professional.

ABSTRACT

It is good to specify in the abstract which parts of the body perceive these vibrations as good.

INTRODUCTION

The first time back pain (low back pain) is mentioned, it should be accompanied by the acronym, because later it appears and one does not know what it is.

Line 48 does not make sense. Reword it differently.

Line 6 reference otherwise. From here on, revise the references because they are out of order.

Line 76, idem to the previous one...it talks about an author without referencing.

Lines 85 and 87 idem

I have the feeling that the second part of the introduction is a copy of another text (because of the different references)...check.

MATERIAL AND METHODS

A weak point is to have an unequal sample between men and women.

Could list the inclusion and exclusion criteria in a more orderly manner (table, order first inclusion and then exclusion...).

Line 129: does not appear as a subtitle. Confusing wording.

Physio therapist is together (line 129).

Line 137: is it another section?

The materials section is confusing. It is not very clear in the exposition of the materials. I recommend revising this part.

RESULTS

Line 228: mentions an author without reference to it. In addition, in the results section, other studies should not be referenced, since what is shown are the results obtained.

Lines 240- 24: belongs to discussion. It makes evaluations regarding the results obtained.

The results shown in question 2 are very subjective, and the statistics are very weak, as well as the description of the results. An adequate statistical analysis should be made for this type of study (qualitative), and not be satisfied with so many per hundreds and average deviation.

DISCUSSION

Quality of movement: in this section there are very few references to previous studies that have analyzed similar proposals. You should back up your assertions with a complete bibliographic search.

Workload: idem to the previous one...it is repetitive because it mentions the results again, when what should be done is to support the results obtained.

Line 349 /361/ 369: reference in a different way.

Lines 381 to 423: repeats data of results. It does not discuss correctly. There are no bibliographic references that refute (or corroborate) the authors' contributions.

References 12, 13, 17, 25...review.

Author Response

(The authors gave the same response as above.)

Round 2

Reviewer 1 Report

Comments and Suggestions for Authors

Thank you for the opportunity to review the manuscript.

Author Response

Thank you for review the manuscript.

Reviewer 2 Report

Comments and Suggestions for Authors

I appreciate the invitation to review again the modifications made with respect to the first submission of the manuscript.
A considerable improvement over the previous submission has been noted.
However, I would still like to make some improvements or proposals:
- lines 73 and 1333 the wording needs to be revised for errors.
- I still think it is a very small sample, so it should be a pilot study.
- Disadvantageous point: questionnaire not validated.
- Results section: It is not understood when quantitative data are shown (3[3;3].  What is shown in square brackets should be clarified, because it gives rise to doubts.

Author Response

Thank you for your helpful review. 
Please find enclosed our comments on the points you have listed.

- The wording in lines 73 and 1333 needs to be checked for errors:
Line 1333 does not exist in the manuscript, we assume that you meant line 133.
We have checked these lines (73 and 133) and adjusted the wording.

- I still think it is a very small sample, so it should be a pilot study:
We have adjusted the title. However, we belief that in this manuscript the term  "feasibility" is more appropriate as an overarching term for pilot studies. We are guided here by Whitehead et al. 2014 https://pubmed.ncbi.nlm.nih.gov/24735841/

- Disadvantageous point: Questionnaire not validated:
We have included this already in the limitations. ("....Another issue is that the questionnaire for physiotherapists has not been validated in previous studies.....")

- Results section: It is not understood when quantitative data are shown (3[3;3].  What is in square brackets should be clarified as it gives rise to doubts.:
In the methods section, we refer to the fact that we work with median and quartile intervals (... "The description of the data is based on the median and the quartiles 0.25 and 0.75....")
However,  now we have taken the above sentence on line 258 out of the manuscript.

We now hope to have found a better solution and thank you very much for your support in improving the manuscript